Leveraging diverse cell-death patterns in diagnosis of sepsis by integrating bioinformatics and machine learning

Liu Mi 1
Gao Xingxing 2
Wang Hongfa 1
Zhang Yiping 1
Li Xiaojun 1
Zhu Renlai 1
Sheng Yunru 1 ty19960316@163.com
1 Center for Rehabilitation Medicine, Department of Anesthesiology, Zhejiang Provincial People’s Hospital, Affiliated People’s Hospital, Hangzhou Medical College , Hang Zhou , China
2 Department of Thyroid Surgery, The First Affiliated Hospital, Zhejiang University School of Medicine , Hang Zhou , China
Uversky Vladimir
Electronic publication date: 2025 Feb 26
Publication date: 2025
Volume: 13
Electronic Location ID: e19077
Received 2024 Nov 4; Accepted 2025 Feb 10
Copyright: © 2025 Liu et al.
Copyright year: 2025
Copyright holder: Liu et al.
License: This is an open access article distributed under the terms of the Creative Commons Attribution License, which permits unrestricted use, distribution, reproduction and adaptation in any medium and for any purpose provided that it is properly attributed. For attribution, the original author(s), title, publication source (PeerJ) and either DOI or URL of the article must be cited.
License URL: https://creativecommons.org/licenses/by/4.0/

Keywords: Sepsis, Programmed cell death, Diagnosis, Machine learning, Nomogram

Funding: The authors received no funding for this work.

==============================
Background

Sepsis is a life-threatening disease causing millions of deaths every year. It has been reported that programmed cell death (PCD) plays a critical role in the development and progression of sepsis, which has the potential to be a diagnosis and prognosis indicator for patient with sepsis.

Methods

Fourteen PCD patterns were analyzed for model construction. Seven transcriptome datasets and a single cell sequencing dataset were collected from the Gene Expression Omnibus database.

Results

A total of 289 PCD-related differentially expressed genes were identified between sepsis patients and healthy individuals. The machine learning algorithm screened three PCD-related genes, NLRC4, TXN and S100A9, as potential biomarkers for sepsis. The area under curve of the diagnostic model reached 100.0% in the training set and 100.0%, 99.9%, 98.9%, 99.5% and 98.6% in five validation sets. Furthermore, we verified the diagnostic genes in sepsis patients from our center via qPCR experiment. Single cell sequencing analysis revealed that NLRC4, TXN and S100A9 were mainly expressed on myeloid/monocytes and dendritic cells. Immune infiltration analysis revealed that multiple immune cells involved in the development of sepsis. Correlation and gene set enrichment analysis (GSEA) analysis revealed that the three biomarkers were significantly associated with immune cells infiltration.

Conclusions

We developed and validated a diagnostic model for sepsis based on three PCD-related genes. Our study might provide potential peripheral blood diagnostic candidate biomarkers for patients with sepsis.

Introduction

Sepsis is a disease associated with immune response dysregulation to infection and severe organ failure (Singer et al., 2016). Due to its high incidence rate and mortality, sepsis is currently a major public health problem (Evans, 2018). The current incidence rate of sepsis is about 3‰ in the United States, which accounts for at least 200,000 deaths each year (Soong & Soni, 2012). The definition of sepsis is the damage to organ function caused by abnormal host responses to infection. At present, the diagnosis of sepsis is mainly based on the sequential organ failure assessment score (SOFA) scores according to the 3.0 definition of sepsis. For patients with infection or suspected infection, sepsis can be diagnosed when the SOFA score increases by ≥2 points from baseline (Singer et al., 2016). However, the SOFA score also has some shortcomings, such as a relatively late diagnostic window and it is a subjective perception-based scoring, which may lead to delayed early treatment of patients (Wang et al., 2023). Nowadays, some biomarkers, including procalcitonin and C-reactive protein, have been applied for the diagnosis and evaluation of sepsis (Mirijello, Tosoni & On Behalf of the Internal Medicine Sepsis Study Group, 2020). Apart from these biomarkers, there are works in various fields such as metabolomics and proteomics that are focused on identifying the biomarkers of sepsis for early diagnosis (Pandey, 2024; Sepsis, 2024). However, in clinical applications, these biomarkers all suffered from insufficient sensitivity and specificity. Due to the lack of early diagnostic biomarkers for sepsis, there is still an urgent need to explore new biomarkers to construct effective models with better sensitivity and specificity.

There are two major ways of cell death according to the triggering mechanism, namely accidental cell death and programmed cell death (PCD). Accident cell death is an uncontrolled biological process, while PCD involves complex rules and various mechanisms. PCD consist of apoptosis, entotic cell death, ferroptosis, netotic cell death, parthanatos, necroptosis, pyroptosis, lysosome-dependent cell death, immunogenic cell death, autophagy-dependent cell death, alkaliptosis and oxeiptosis (Tang et al., 2019). The latest researches have also identified two novel ways of cell death, named cuproptosis and disulfidptosis (Tsvetkov et al., 2022; Liu et al., 2023). PCD is closely related to the occurrence and development of various diseases including sepsis. Apoptosis is a cell death process controlled by certain genes, characterized by autonomy and orderliness, in order to maintain a stable internal environment (Xu, Lai & Hua, 2019). When apoptosis is blocked, necroptosis is activated by the extracellular signal or intracellular signal and resulted in the self-destruction of cells. Necroptosis is associated with various inflammatory diseases and is involved in the process of resisting microbial or viral invasion (Yan et al., 2022). Netotic cell death is associated to with the occurrence and development of multiple organ failure in sepsis patients (Brown et al., 2006). Pyroptosis plays an important role in the imbalance of hemostasis and immune thrombosis formation in sepsis (Zhu et al., 2023). Ferroptosis is a double-edged sword in sepsis, as it promotes bacterial invasion and sepsis induction, while also helping immune cells clear pathogens (Amaral et al., 2019). Oxeiptosis, the antioxidant response and reactive oxygen species-induced cell death, involving inflammation in sepsis (Oikawa et al., 2022).

Studies have attempted to improve molecular diagnosis and prognosis based on high-throughput sequencing such as transcriptome, proteome, or metabolic profiling (Reinhart et al., 2012). Previous studies have attempted to screen biomarkers for the diagnosis of sepsis through PCD related genes. Liang et al. (2022) developed a risk score system for the prognosis of sepsis patients through univariate Cox analysis and least absolute shrinkage and selection operator (LASSO) Cox regression analysis based on six pyroptosis-related genes. Wang et al. (2023) screened out ten PCD-related genes for diagnosis of sepsis from 11 PCD patterns.

In this study, we aimed to construct a high-precision diagnostic model and identify potential therapeutic targets for sepsis management. Specifically, we analyzed the differential expression of 14 PCD patterns related genes. Based on three machine learning algorithms, we screened three potential biomarkers (NLRC4, S100A9 and TXN) and constructed a diagnostic model for sepsis. Finally, we validated the differential expression of the three biomarkers in sepsis patients and healthy individuals from our center.

Materials and methods

PCD-related genes collect

PCD-related genes were collected from public gene sets and manual collation (Tang et al., 2019). Finally, 1,089 PCD-related genes were selected in this study, including 580 apoptosis genes, 52 pyroptosis genes, 88 ferroptosis genes, 367 autophagy genes, 15 entotic cell death genes, 101 necroptosis genes, 100 immunogenic cell death genes, nine parthanatos genes, eight netotic cell death genes, seven alkaliptosis genes, death genes, 220 lysosome-dependent cell and five oxeiptosis genes, 14 cuproptosis genes and 91 disulfidptosis genes (Table S1).

Datasets collect

The microarray data and clinical features of sepsis patients were acquired from the Gene Expression Omnibus database (GSE95233, GSE154918, GSE28750, GSE57065, GSE13904, GSE26378 and GSE26440) (Venet et al., 2017; Herwanto et al., 2021; Sutherland et al., 2011; Tabone et al., 2019; Wong et al., 2009; Grunwell et al., 2015; Wong et al., 2009). “AnnoProbe” R package was used for mapping of the probes. “Limma” R package was used for calculating the average values of multiple probes (Ritchie et al., 2015). GSE95233 and GSE154918 were used as the analysis set, GSE28750, GSE57065, GSE13904, GSE26378 and GSE26440 were used as the validation set.

Differential analysis

Gene expression data was normalized corrected using the “sva” package (Leek et al., 2012). This was done after GSE95233 and GSE154918 were combined through the Combat algorithm. The assessment of batch-effect removal was analyzed by PCA (Fig. S1). Differential expression genes (DEGs) between the sepsis and the healthy control group was identified using the “LIMMA” package. We selected |log2FC| > 0.5 and adjusted P value < 0.05 as the cutoffs for DEGs.

Functional enrichment analysis

GO and KEGG pathway enrichment analyses were performed using the “clusterProfiler” package in R software. The ssGSEA algorithm was employed to analyze different programmed cell death patterns between the sepsis and control (Hänzelmann, Castelo & Guinney, 2013).

Screening of the diagnostic markers

Random forests (RF), LASSO logistic regression and SVM-RFE were used for biomarker selection using “randomForest”, “glmnet” and “e1071” R packages (Tai et al., 2019; Wang, Yang & Luo, 2016; Ishwaran & Kogalur, 2010). The number of the trees in Ramdom forest models was 100. The kernel type for SVM-RFE was linear. The penalty parameter in LASSO was lambda.min. Then we select genes that existed in all three models for further research.

Nomogram construction for diagnosis of sepsis

The nomogram that combined three potential biomarkers was built using the rms package. The calibration curve was used to evaluate the performance of the nomogram.

Immune cell evaluation

We evaluated 22 types of immune cells by the website CIBERSORT. The spearman correlation between biomarkers and immune cells was calculated.

Gene set enrichment analysis

To explore the biological function of three biomarkers, gene set enrichment analysis (GSEA) analysis was executed in low- and high-expression groups (Subramanian et al., 2005). KEGG and GO gene sets were selected. The normalized enrichment score and false discovery rate was used to assess the significance. The top ten significant KEGG pathway or biological process were selected.

scRNA sequencing dataset download and analysis

The single cell dataset GSE167363 (including two healthy individuals and three sepsis patients) was downloaded from the GEO database. The sequencing data was processed by R package “Seurat.” Cells with gene counts between 200 and 4,000 and UMI counts below 30,000 were selected, and finally 29,633 cells were selected for further analysis. The Harmony method was applied for batch correction. We selected 2,000 variable features and separated these cells to 16 clusters. Seven main cell types were identified based on the classic markers.

qRT-PCR

Total RNA was extracted from 200 μL blood samples of five sepsis patients and five healthy individuals collected from Zhejiang Province Hospital through macrodissection according to the protocol of blood total RNA purification kit (Sangon Biotech, Shanghai, China). All study participants were 18 years or older, had a body mass index <40, and provided written informed consent. The screening criteria for sepsis patients were based on Sepsis 3.0, and had a clinical suspicion of systemic infection based on microbiological diagnoses. Potential sepsis participants admitted to the ICU and, patients admitted for planned major open surgery, were excluded from the study if they had any systemic immunological disorders including systemic lupus erythromatosus, crohn’s disease and insulin-dependent diabetes mellitus (type 1 diabetes) or were transplant recipients or currently receiving chemotherapy treatment for cancer. Samples were stored in −80 °C and used for further experiments in a week. The DEPC water was used for purifying all the equipments involved in this experiment. The RNA was dilution with the RNAse free water. We remove DNA and purify RNA through the adsorption column in this kit. We evaluated the quality of RNA through the A260/A230 and A260/A280 value of each sample by Multiskan Skyhigh (Thermo Fisher Scientific, Waltham, MA, USA). Then we conformed all the values to the standard. RNA (200 ng for each sample) was reverse-transcribed into cDNA by reverse transcriptase and random primer (N9) (0.1 μg/μL) in a 20 μL reaction volume with transcript one-step gDNA removal and cDNA synthesis supermix kit (TRANSGEN, Beijing, China). The reverse-transcribed reaction conditions included 25 °C for 10 min, 42 °C for 15 min and 85 °C for 5 s. cDNA was stored in −20 °C and used for further experiments in a week. RT-qPCR was performed through 2 × SYBR Green qPCR Mix (Sparkjade). QuantStudio Design & Analysis Software (ABI, Waltham, MA, USA) was used for manual reaction setup and analysis. The PCR conditions included a denaturing step at 94 °C for 30 s and 45 cycles of 94 °C for 5 s and 60 °C for 30 s for real time plate read. The melt curve stage included 95 °C for 15 s, 60 °C for 1 min and 95 °C for 1 s. The whole reaction volume was 20 μL, and the concentration of forward and reverse primers were 10 μM. SYBR Green I was the fluorescent dye applied in this assay. ACTB was used for internal controls. The 2−ΔΔCt value was quantitatively calculated as the gene expression. The RT-qPCR was repeated three times. The results showed well repeatability. Each melt curve has only one peak and we found little outliers. Detailed primer sequences were described in Table 1. Sangon Biotech (Shanghai, China) designed all primers. The studies involving human participants were reviewed and approved by Clinical Research Ethics Committee of the Zhejiang Provincial People’s Hospital (2024-225). Written informed consent was obtained from all subjects or their legal guardian.

Table 1 List of the primers in this study.

Gene	Species		Sequence	
ACTB	Human	Forward	CATGTACGTTGCTATCCAGGC	
Reverse	CTCCTTAATGTCACGCACGAT	
NLRC4	Human	Forward	TCAGAAGGAGACTTGGACGAT	
Reverse	GGAGGCCATTCAGGGTCAG	
S100A9	Human	Forward	GGTCATAGAACACATCATGGAGG	
Reverse	GGCCTGGCTTATGGTGGTG	
TXN	Human	Forward	GTGAAGCAGATCGAGAGCAAG	
Reverse	CGTGGCTGAGAAGTCAACTACTA	

Results

Workflow of this study

For training and validation cohorts, we identified 73 individuals from the GSE95233 dataset (51 sepsis patients and 22 healthy individuals), 79 individuals from GSE154918 (39 sepsis patients and healthy individuals), 30 individuals from GSE28750, 53 individuals from GSE57065, 117 individuals from the GSE13904, 103 individuals from the GSE26378 and 140 individuals from the GSE26440. We merged GSE95233 and GSE154918 as the training set, the others as validation set. We also downloaded the single cell sequencing data from GSE167363 (two healthy individuals and three sepsis patients). Besides, 14 PCD patterns with 1,089 concatenated genes were selected for further analysis (Table S1). The flow diagram of the study was showed in the Fig. 1.

Figure 1 Flowchart for analysis of programmed cell death patterns in sepsis patients.

Landscape of PCD-related genes in sepsis patients

In the merging dataset, we identified 289 DEGs, 180 of which were upregulated and the other 109 genes were downregulated in the sepsis group (Figs. 2A and 2B, Table S2). The results of GO enrichment analysis showed that these DEGs were involved in multiple pathways, including TNF, mTOR, NF-kappa B and Chemokine signaling pathways, etc. (Figs. 2C and 2D). Moreover, we calculated the scores of each programmed cell death using ssGSEA algorithm. The enrichment score of eight programmed cell death patterns (alkaliptosis, autophagy, entotic cell death, ferroptosis, lysosome-dependent cell death, netotic cell death, parthanatos, pyroptosis) were upregulated in sepsis group and three patterns (disulfidptosis, necroptosis, oxeiptosis) were downregulated (Fig. 2E).

Figure 2 (A–E) Landscape of PCD-related genes in sepsis patients.

Heatmap. ** p < 0.01; **** p < 0.0001.

Screening of the diagnostic biomarkers

By comparing the overlapping of the DEGs and programmed cell death related genes, we identified 289 overlapping genes (Fig. 3A). We applied thee machine learning algorithms to identify potential biomarkers: Random Forest, SVM-RFE and LASSO regression (Table S3). For SVM-RFE model, the cross-validation error of the model reached the minimum while the selected gene number is 28 (Fig. 3B). The model based on the 28 genes achieved an AUC value of 0.94 ± 0.04 under five-fold cross validation (Fig. 3C). For Random Forest model, the cross-validation error of the model reached the minimum while the selected gene number is 12 (Figs. 3D and 3E). The model based on the 12 genes achieved an AUC value of 1.00 ± 0.00 under five-fold cross validation (Fig. 3F). The LASSO regression analysis identified 17 genes among the statistically significant univariate variables with the average AUC of 0.99 ± 0.01 (Figs. 3G–3I).

Figure 3 (A–I) Detection of diagnostic markers by three machine learning methods.

Validation of the diagnostic markers

By Venn diagram, we identified NLRC4, S100A9 and TXN as genes with overlapping regions (Fig. 4A). The expression of NLRC4, S100A9 and TXN was significantly higher in the sepsis patients (Fig. 4B). We further developed a nomogram model based on three biomarkers (Fig. 4C). The decision curve analysis indicated that the nomogram model offered a good clinical benefit (Fig. 4D). The ROC curves for this model showed performance with an AUC of 100.0% (Fig. 4E). In the validation datasets of GSE28750, GSE57065, GSE13904, GSE26378 and GSE26440, the expression of NLRC4, S100A9 and TXN were also significantly higher in the sepsis group (Figs. 5A, 5C, 5E, 5G and 5I). With AUCs of 100.0%, 99.9%, 98.9%, 99.5% and 98.6% respectively (Figs. 5B, 5D, 5F, 5H and 5J).

Figure 4 (A–E) The model constructed for diagnosis of sepsis.

**** p < 0.0001.

Figure 5 (A–J) Validation of the model for diagnosis of sepsis.

**** p < 0.0001.

Infiltration of immune cells in sepsis patients

By applying the CIBERSORT algorithm, sepsis patients showed a higher proportion of neutrophil, memory activated CD4+ T cells, naive CD4+ cells, gamma delta T cells, M0 macrophages, activated myeloid dendritic cell. The proportion of CD8+ T cells, resting NK cells resting CD4+ memory T cells and activated NK cells were relatively lower (P < 0.05) (Figs. 6A and 6B). The correlation analysis showed that NLRC4, S100A9 and TXN exhibited a significant correlation with various immune cells (Fig. 6C).

Figure 6 (A–C) Immune cell infiltration between sepsis and healthy patients.

** p < 0.01; **** p < 0.0001.

GSEA analysis of the potential biomarkers

The three potential biomarkers (NLRC4, S100A9 and TXN) were subjected to GSEA analysis to investigate their biological functions (Fig. 7). The high expressed genes (NLRC4, S100A9 and TXN) were mainly enriched in golgi vesicle transport, macrophage activation, oxidative phosphorylation. And the low expressed genes (NLRC4, S100A9 and TXN) were enriched in immune response associated pathways, like antigen processing and presentation, t cell receptor signaling pathway and natural killer cell mediated immunity. These results suggested that these gene might involve in various pathways that contributed to progression of sepsis.

Figure 7 (A–F) Single-gene GSEA analysis of three biomarkers.

Validation of the diagnostic markers in single-cell sequencing data

To further investigate the expression level of these diagnostic genes in single cell level, we downloaded the single cell sequencing data from GSE167363. After dimension reduction and clustering, we selected 2,000 variable genes and separated these cells to 16 clusters. Seven main cell types (T cells, B cells, myeloid/monocytes, NK cells, dendritic cells, megakaryocytes and red blood cells) were identified based on the classic markers (Figs. 8A–8C). Myeloid/monocytes, T cells and B cells were the main cell types in both healthy control and sepsis group (Fig. 8D). The markers of each cell group were showed in the TSNE feature plot (Fig. 8E). Then we explored the expression of three diagnostic genes in different cell types. The expression of S100A9, NLRC4 and TXN were specifically expressed in myeloid/monocytes and dendritic cells (Figs. 9A and 9B). Moreover, the expression of S100A9, NLRC4 and TXN were significantly higher in sepsis group (Figs. 9C and 9D).

Figure 8 (A–E) Identifying the main cell types in sepsis and healthy individuals.

Figure 9 (A–D) Cellular localization of three diagnostic markers in scRNA sequencing data.

**** p < 0.0001.

Expression validation of diagnosis-related genes

Finally, we performed qRT-PCR to validate the expression levels of the three biomarkers. The whole blood samples from five sepsis patients and five healthy individuals were collected from Zhejiang Province Hospital. The results showed that the expression of NLRC4, S100A9 and TXN was significantly higher in sepsis patients (Fig. 10).

Figure 10 (A–C) Transcriptional expression in whole blood from sepsis patients and healthy controls of NLRC4.

** p < 0.01; *** p < 0.001.

Discussion

Overall, this study comprehensively analyzed fourteen programmed cell death patterns, screened out three biomarkers for sepsis through three machine learning algorithms, and further validated their excellent performance in five other external cohorts. A nomogram including the three biomarkers was constructed, and the results indicated the good performance of this model. Furthermore, we validated the effectiveness of these biomarkers in clinical samples from our center.

Increasing evidence suggested that programmed cell death played a critical role in sepsis (Yang, Coopersmith & Lyons, 2024). Sepsis represents a condition characterized by a widespread inflammatory response triggered by infection, with immune dysregulation serving as its fundamental mechanism. Central to the physiological disturbances in sepsis are endothelial malfunction and the degranulation of neutrophils (Martín-Fernández et al., 2021). Initially, sepsis patients exhibit a heightened inflammatory state, which subsequently transitions into a prolonged immunosuppressive phase (Hotchkiss, Monneret & Payen, 2013). During sepsis, apoptosis-induced cell death contributes directly to microvascular dysfunction and organ failure, while apoptotic immune cells may predispose patients to secondary infections or further immune suppression (Cao, Yu & Chai, 2019). Moreover, autophagy dysfunction triggers CD4+ T-cell apoptosis in sepsis. Wang et al. (2022) discovered that the deletion of mTOR can alleviate this symptom by enhancing autophagy-lysosomal fusion.

Our study revealed a total of 289 DEGs in PCD-related genes between sepsis patients and healthy individuals, with 180 genes upregulated and 109 genes downregulated. Function enrichment analysis indicated that the DEGs were mainly associated with the immune response-regulating signaling pathway, including TNF, mTOR, NF-kappa B and Chemokine signaling pathways. On the basis of these DEGs, we applied three machine learning algorithms to screen and identify potential biomarkers for sepsis diagnosis. We screened a signature featuring three programmed cell death related genes (NLRC4, TXN and S100A9) and found that it could predict the occurrence of sepsis. NLRC4 encodes the nucleator of inflammasome (Bauer & Rauch, 2020). NLRC4, a crucial constituent of the inflammasome, plays a role in endogenous danger signaling in response to various microbial stimuli and macrophage activation (Kofoed & Vance, 2012). The activation of the NLRC4 inflammasome could significantly impact gram-negative bacterial infections, particularly those linked to Salmonella typhimurium (Sundaram & Kanneganti, 2021). Studies have indicated that elevated levels of NLRC4 enhance macrophage inflammasome function, potentially contributing to conditions such as infantile small bowel colitis syndrome and recurrent macrophage activation syndrome (Canna et al., 2014). Upon detecting danger signals or pathogen-associated molecular patterns, the NLRC4 sensor triggers the activation of caspase-1, representing a common mechanism for inflammasome vesicle activation (Man, Karki & Kanneganti, 2017). In this study, we found that NLRC4 was upregulated in patient with sepsis. Hence, it is plausible to speculate that NLRC4 may induce cell death through caspase-1 activation and enhancement of the inflammatory response, ultimately contributing to the onset of sepsis. Thioredoxin (TXN) plays a pivotal role in the immune response and redox processes as a key antioxidant system dependent on mercaptans. It contributes to the body’s defense against oxidative stress by regulating cellular free radicals and reactive oxygen species, thereby potentially preventing cell death. Furthermore, TXN is considered essential in inflammatory processes (Zheng & Conrad, 2020). The upregulation of TXN has been verified to be significantly associated with inflammation in sepsis patients (Shao et al., 2020). Our results also indicated that TXN was a potential biomarker for sepsis. S100A9 encodes a calcium-binding proteins and form a heterodimer which is known as calprotectin with S100A8. S100A9 is related to the activation of inflammation and the immune system (Cerón et al., 2023). S100A9 is upregulated in sepsis patient’s blood sample and is associated with various complications of sepsis, including sepsis-induced acute liver injury (Zhang et al., 2023), sepsis-induced myocardial dysfunction (Jakobsson et al., 2023) and lung damage (Ding et al., 2021). Dai et al. (2017) found that intracellular S100A9 promotes myeloid-derived suppressor cells in late sepsis. Zhang et al. (2023) found that S100A9 plays an essential pro-inflammatory role in sepsis-mediated acute liver injury by regulating AKT-AMPK-dependent mitochondrial energy metabolism.

The CIBERSORT analysis was performed in this study to analyze the immune cell infiltration between sepsis and control groups. Dynamic changes of various immune cells could be associated with the incidence and development of sepsis. NK cells constitute the first line of defense against pathogen invasion (Ma et al., 2021). It has been reported that sepsis patients showed a significantly lower number of NK cells than patients with community-acquired pneumonia (CAP) (Forel et al., 2012). The lower level of NK cells indicated the decrease of survival rate of sepsis patients (Giamarellos-Bourboulis et al., 2006). Classically activated macrophages reprogrammed into an immunosuppressive phenotype after the initial stage of inflammation in sepsis (Patoli et al., 2020). Furthermore, increasing studies have revealed that neutrophils were necessary for the control of pathogens in the early stage of sepsis (Liu & Sun, 2019).

The biomarkers identified in our research hold promise for potential utilization in the early detection of sepsis. Utilizing these biomarkers for early detection can facilitate prompt therapeutic measures, potentially decreasing the incidence of morbidity and mortality linked to delayed diagnosis. Furthermore, these biomarkers could be used in tandem with existing diagnostic methods (such as procalcitonin and C-reactive protein) to enhance accuracy and reduce false-positive rates. The identification of these markers for sepsis through machine learning model might pave the way for more targeted interventions. Although initial gene expression analysis may incur higher costs compared to traditional methods, the long-term benefits, including accelerated diagnosis, tailored treatments, and potentially shorter hospital stays, could ultimately yield greater cost-effectiveness. We envision our model not as a replacement but as a complementary tool alongside existing diagnostic methods. While our model offers rapid insights, existing diagnostic methods remain invaluable for their definitive evidence of sepsis and for guiding therapeutic choices.

Although the diagnostic model exhibited a good performance in training and testing datasets, there are still several limitations. Firstly, the datasets used in this study are all retrospective cohorts, which may lead to certain biases. Although we have validated the upregulation of NLRC4, TXN, and S100A9 expression in the peripheral blood of sepsis patients through qRT-PCR, there are still some deficiencies, such as limited number of enrolled patients and the lack of multicenter validation. Furthermore, the biological function of several biomarkers such as TXN in our model has not been investigated in cell and mouse model, which is essential for further experimental research.

Conclusions

In conclusion, this research proposed three programmed cell death related genes (NLRC4, TXN, S100A9) as practical biomarkers for sepsis patients. The diagnostic model constructed in this study might make a notable difference in the early identification of sepsis patients.

Supplemental Information

Supplemental Information 1 Supplemental tables.

Supplemental Information 2 Supplemental figures.

Supplemental Information 3 MIQE checklist.

Supplemental Information 4 Raw qPCR data.

Additional Information and Declarations

Competing Interests

The authors declare that they have no competing interests.

Author Contributions

Mi Liu conceived and designed the experiments, authored or reviewed drafts of the article, and approved the final draft.

Xingxing Gao analyzed the data, prepared figures and/or tables, and approved the final draft.

Hongfa Wang performed the experiments, prepared figures and/or tables, and approved the final draft.

Yiping Zhang performed the experiments, analyzed the data, authored or reviewed drafts of the article, and approved the final draft.

Xiaojun Li performed the experiments, analyzed the data, authored or reviewed drafts of the article, and approved the final draft.

Renlai Zhu performed the experiments, authored or reviewed drafts of the article, and approved the final draft.

Yunru Sheng conceived and designed the experiments, analyzed the data, prepared figures and/or tables, and approved the final draft.

Human Ethics

The following information was supplied relating to ethical approvals (i.e., approving body and any reference numbers):

The studies involving human participants were reviewed and approved by Clinical Research Ethics Committee of the Zhejiang Provincial People’s Hospital (2024225).

Data Availability

The following information was supplied regarding data availability:

The raw data from the qRT-PCR experiments are available in the Supplemental Files.

Sequences are available at GEO: GSE95233, GSE154918, GSE28750, GSE57065, GSE13904, GSE26378 and GSE26440.

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
