# Peer review of "Leveraging diverse cell-death patterns in diagnosis of sepsis by integrating bioinformatics and machine learning"

_PeerJ, doi:10.7717/peerj.19077_

## Round 0.1 · original submission · Major Revisions

Please address concerns of all reviewers and revise manuscript accordingly.

Reviewer 1 ·

Basic reporting

Apart from few suggestion enlisted below the reporting is fine

Experimental design

Apart from a few suggestion enlisted below it is fine

Validity of the findings

Apart from a few suggestion enlisted below it is fine

Additional comments

The author mentions ”At present, the diagnosis of sepsis is mainly based on the sequential organ failure assessment score (SOFA) scores according to the 3.0 definition of sepsis”.
They need to elaborate on the definition of sepsis according to Consensus 3.0 and cite the publication here.
The author can add more shortcomings of utilizing the SOFA score such as it is a subjective perception-based scoring that may affect the SOFA scoring.
Apart from PCT, there are works in various fields such as metabolomics (https://doi.org/10.15441/ceem.24.211) (10.7150/ntno.94071) and proteomics that are focused on identifying the biomarkers of sepsis for early diagnosis. The authors need to mention the ongoing omics work for biomarker identification as well.
“Due to the lack of early diagnostic biomarkers for sepsis, there is still an urgent need to explore new biomarkers to construct effective models with better sensitivity and specificity”. The author should refrain from mentioning the lack of early diagnostic biomarkers. As mentioned previously there are various omics work in this direction.
Toward the end the author mentioned omics but they need to be mentioned in the being. Apart mentioned metabolomics and proteomics, the author can talk about utilizing genes. Then talk about the PCD gene of interest in this manuscript.
In the methods section for qRT PCR. The author needs to mention the details of the sepsis patient and healthy control such as the inclusion and the exclusion criteria for the patients.
The discussion needs to be elaborated. The final set of genes and their correlation or the pathways that are affected in sepsis needs to be explained.

Reviewer 2 ·

Basic reporting

The study focuses on leveraging transcriptome data related to various patterns of programmed cell death (PCD) to develop a diagnostic model for sepsis. The authors integrated multiple microarray datasets, validated the model with external cohorts, and performed single-cell RNA sequencing (scRNA-seq) analysis, ultimately identifying three PCD-related biomarkers (NLRC4, S100A9, and TXN). While the approach is innovative and potentially valuable for early sepsis diagnosis, there are several points that merit attention to strengthen the manuscript’s rigor, clarity, and impact.

Experimental design

Major Points
1. Data Integration and Batch-Effect Correction Combining GSE95233 and GSE154918 as the training set. It is stated that the sva package was used for normalization and batch correction. However, the specific procedure (e.g., ComBat, other sva methods) and the visualization/assessment of batch-effect removal (e.g., PCA or UMAP plots before and after correction) are not described.
How were the platform differences and other potential confounding factors (e.g., patient demographics, RNA quality) handled?

2.Consistency and Comparability Among Datasets. While the authors merged some datasets and used others for validation, it is crucial to show that the merged training set became sufficiently homogeneous to allow reliable downstream analysis. If multiple probes mapped to a single gene, were those merged simply by averaging expression values, or was a more sophisticated approach taken?

3. Sample Size and Class Imbalance
1)Training Set Composition
The manuscript mentions 51 sepsis patients and 22 controls in one dataset, and 39 sepsis patients and 40 controls in another, but it is unclear if these samples are balanced once merged.
Class imbalance (sepsis vs. control) can lead to overfitting or skewed results in machine learning models.

2)External Validation Cohorts
Several external datasets are used (GSE28750, GSE57065, etc.), each with distinct sample sizes. Were there any measures taken to ensure these sets are representative and properly balanced?

3)Clinical qPCR Validation
Only five sepsis samples and five healthy samples were tested in the qPCR experiment. This small sample size may limit the power of the validation and the generalizability of the findings.

4. Thresholds for Differential Expression
1) Cutoff (|log2 FC|> 0.5and adjusted p < 0.05)
A |log2 FC| threshold of 0.5 is relatively lenient and may yield a large gene set with modest fold changes.How did the authors ensure that such a threshold effectively captures clinically meaningful differences rather than subtle, possibly non-specific changes?

2)Biological vs. Statistical Significance
While applying standard DE analysis procedures is valid, some additional biology-driven filters may be helpful (e.g., prior evidence linking genes to sepsis or to the specific PCD pathways).
Clarifying these points could strengthen the confidence in the final gene list.

5.Machine Learning Approaches and Feature Selection
Use of Random Forest, SVM-RFE, and LASSO
1.The three methods are appropriate for feature selection, but the manuscript does not thoroughly describe hyperparameter tuning (e.g., number of trees in RF, penalty parameter in LASSO, kernel type for SVM).
2.Was cross-validation or grid-search performed for parameter optimization?
Intersection of Selected Genes
1.The authors ultimately take the intersection of genes identified by all three methods and arrive at three biomarkers. This is a strict approach but might exclude potentially important features identified by only one method.
2.Why was the intersection chosen instead of a union or ensemble approach? How were the final three genes prioritized beyond simply being present in all three models?
Model Validation and Comparisons
1.The paper reports excellent AUCs (approaching or reaching 1.0) in both training and validation sets. While this is promising, extremely high AUCs raise the possibility of overfitting or data leakage, especially in retrospective studies.
2.A more comprehensive breakdown of sensitivity, specificity, PPV, and NPV would help convey the potential clinical utility more concretely.
6. Clinical Validation and Utility
Limited Clinical Samples
Validation in only five sepsis and five control samples via qPCR is a start, but it is too small a cohort to establish robust diagnostic performance. Additional independent clinical cohorts are necessary.
Clarify the inclusion/exclusion criteria for these five sepsis patients (e.g., underlying infection type, disease severity) to demonstrate the representativeness of the sample.

Diagnostic Cutoff and Clinical Workflow
The study shows promising results for using NLRC4, S100A9, and TXN in a nomogram. However, it would be helpful to propose a clinically relevant cutoff or combination score that could be tested prospectively for early sepsis detection.
A discussion comparing these novel biomarkers to existing clinical markers (e.g., PCT, CRP) or scores (e.g., SOFA) would also illustrate potential advantages or limitations in real-world settings.

7. Mechanistic Insight and Literature Context
PCD Pathways Involvement
The authors list 14 PCD modes and identify key genes, but the in-depth mechanistic link among these pathways, the three key genes, and sepsis progression could be elaborated further.
For instance, how do NLRC4 (inflammasome-related), TXN (redox regulation), and S100A9 (calprotectin complex) interact or converge in immune dysregulation?
Functional Studies
While bioinformatic and expression-level findings suggest these biomarkers’ diagnostic potential, further mechanistic (in vitro or in vivo) experiments would help confirm their causal roles and address whether upregulation is a driver of pathology or a byproduct of inflammation.

Validity of the findings

no comment'

---

## Round 0.2 · accepted · Accept

All concerns of the reviewers were addressed and revised manuscript is acceptable now.

Reviewer 1 ·

Basic reporting

The author have addressed all the queries

Experimental design

The author have addressed all the queries

Validity of the findings

The author have addressed all the queries

Additional comments

The author have addressed all the queries

Reviewer 2 ·

Basic reporting

yes

Experimental design

yes

Validity of the findings

yes

Additional comments

no